# Comparative Effectiveness of Artificial Intelligence-Based Interactive Home Exercise Applications in Adolescents with Obesity

**DOI:** 10.3390/s22197352

**Published:** 2022-09-28

**Authors:** Wonjun Oh, Yeongsang An, Seunghwa Min, Chanhee Park

**Affiliations:** 1Department of Physical Therapy, Yonsei University, Wonju 26493, Korea; 2Funrehab Co., Ltd., Daejeon 35229, Korea

**Keywords:** artificial intelligence, home exercise, obesity, adolescent, overweight

## Abstract

The rate of obesity in adolescents has increased due to social distancing measures and school closures caused by the COVID-19 pandemic. These issues have caused adolescents to change their lifestyles and eating habits. Furthermore, the growth in inactive behavior and computer screen or watching TV time, as well as the reduction in physical activity, could similarly be related with obesity. To overcome this problem, we recently developed an artificial intelligence (AI)-based gesture recognition game application called Super Kids Adventure (SUKIA, Funrehab, Daejeon, Korea), which provides inexpensive and motivational game applications. This research is designed to assess the effects of SUKIA and Nintendo Switch (NINS) on calorie consumption, VO2 max, 6-minute walking test (6MWT) as well as body mass index (BMI), and the Borg rating of perceived exertion scale (RPE) in adolescents with obesity. A convenience sample of 24 adolescents with obesity were randomized into either the NINS or SUKIA groups 5 days/week for 3 weeks. Analysis of variance (ANOVA) and independent *t*-tests were presented with significant level at *p* < 0.05, and the analysis indicated that SUKIA showed superior effects on calorie consumption, VO2 max, and RPE compared to NINS. Our results provide evidence that SUKIA can more effectively improve cardiopulmonary function and calorie consumption than NINS in adolescents with obesity during COVID-19.

## 1. Introduction

Globally, obesity, or the percentage of the population that is overweight, has become more widespread and has yielded major public health concerns in children and adolescents. In the period 2017–2018, the global incidence of adolescents with obesity (overweight) was 19.3% and impacted about 14.4 million adolescents [1]. Obesity is a acknowledged risk factor for abundant health difficulties, containing cardiovascular diseases, hypertension, diabetes, high cholesterol as well as respiratory complications (asthma), musculoskeletal illnesses (arthritis), and some systems of cancer, and death also increase gradually formerly the obesity threshold is intersected [1]. In reply to COVID-19, international measures were executed by governments, involving school closures, lockdowns, recommendations for social distancing, and other measures to mitigate virus spread, to decrease the burden on healthcare procedures [2]. These considerations led adolescents to change their lifestyles and eating habits. Autonomously, the growth in inactive behavior and smartphone and TV screen period, as well as the reduction in physical activity, which could additionally be related with obesity [3,4]. As has been reported, adolescents were likely to gain weight throughout vacations or holidays, and it has been argued that adolescent obesity rates might expand proportionally to the number of months that schools are shut down [5]. Statements from the US indicated that 1.27 million additional case of childhood obesity were noted in 2020 [5]. Consequently, the necessity to encourage the implementation of a healthy nutritious condition in adolescents throughout the COVID-19 virus has been enthusiastically encouraged. Current obesity prevention approaches, including conventional high-intensity interval training, have been widely used to address body mass and cardiometabolic risks in adolescents with obesity [6,7]. However, due to COVID-19, exercise group recruitment and group exercise have become difficult. There is a clear need for more research on effective and innovative exercise programs for adolescents with obesity under COVID-19 pandemic circumstances [4]. Moreover, adolescents are repeatedly bored with and disconnected from the conventional exercise program, which may undesirably move the conventional exercise program’s efficiency and motivation. It is then supposed that game-based exercise could answer this and be an applicable system to inform adolescent about advertisement [8].

Given the small amount of activity possible due to the COVID-19 pandemic, it is essential to develop innovative and effective physical activity intervention programs that increase activity at home and improve adolescent’s health.

Available gaming machines promote better activity as well as interaction throughout gaming by encouraging total body motion movement through the application of wireless hand-held controllers (such as the Nintendo Wii, Nintendo Co. Ltd., Minami-ku, Kyoto, Japan) [9,10]. Even though video games using wireless hand-held controllers have been stated to meaningfully increase adolescents’ body movement and calorie expenditure associated with inactive video games, no experimental study has documented the influence of body motion movement to energy expenditure throughout dynamic video gaming [9,11]. The Nintendo Wii provides a well-programmed system to generate motivation. However, it is expensive and not readily affordable to many.

One innovative intervention strategy that has shown promise in the health care field is artificial intelligence (AI)-based gesture recognition interventions. We recently developed an AI-based gesture recognition game application called Super Kids Adventure (SUKIA, Funrehab, Daejeon, Korea), which provides inexpensive and motivational game applications. In essence, the SUKIA program is designed to make it easy and fun for users to play games and exercise.

This study compares the effects of SUKIA and the Nintendo Switch (NINS) on calorie consumption, VO2 max, the 6-minute walking test (6MWT) as well as body mass index (BMI), and Borg rating of perceived exertion scale (RPE) in adolescents with obesity. Our hypothesis is that there would be differences in calorie consumption, VO2 max, 6MWT, BMI, and RPE between the NINS and SUKIA.

## 2. Methods

### 2.1. Participants

A convenience sample of 24 adolescents (4 females; mean age: 13.2 ± 3.6) in middle school and high school was enrolled. Caregivers and participants had written assent and informed consent, individually. Consequently, patient consent was not required. This study was approved by the community welfare center internal review board (IRB No. IRB-2022-01). The inclusion criteria were: (1) age ranging from 10 to 17 years; (2) BMI over the 85% (overweight); (3) no pharmacologic treatment; (4) no sign of hormonal, orthopedic, as well as metabolic and cardiovascular illness at the time of the research’s commencement; and (5) no involvement in any even implementation training program (excluding physical schooling programs 2 days a week) for a minimum 6 months prior to the origination of the study and through the protocol.

### 2.2. Outcome Measurements

Calorie consumption was used to determine how much exercise was performed. The metabolic equivalent (MET) is an indicator of how much more difficult it is to perform an activity with the amount of oxygen required for the body to function at a stable level (1 MET). In this study, METs presented in the Guidelines of the American College of Sports Medicine (ACSM) were used [12]. Table 1 shows the MET values according to physical activity and speed [13,14].

Calorie consumption was calculated using Equation (1) to calculate the metabolic equivalent for each speed according to each behavior.
(1)$$Calorie\;consumption=1.05\times METs\times \frac{Time}{3600}\times Weight$$

Time is the exercise time (s), weight is shown as kg, and 1.05 is the constant for calculating calorie consumption [13].

VO2 max was used to determine measurement of aerobic fitness. VO2 max was measured as the maximum rate at which one can take oxygen from the air and deliver it through the lungs into the bloodstream for use by working muscles. Participants were requested to come in 3 h following their lunch. They were instructed not to engage in any type of active training 48 h in advance of the test. The Queens College Step Test was applied to calculate maximum aerobic capacity. It is a typical technique to calculate one’s maximum oxygen comprehension applying submaximal workout into the way of bench stepping, which was appropriate for participants. Before the test, participants were requested to loosen up with 5 min of vigorous stretching and walking of the lower extremity muscles. A 16¼ inch stepping bench was utilized together with a metronome and stopwatch. The metronome was used to observe the stepping cadence, which was set at 96 beats per minute (24 complete steps per min). The step test commenced following a briefing presentation and training phase. The participants were expected to execute each one stepping cycle to a four-step cadence, up, up and down, down constantly for 3 min. Following achievement of the experiment, participants continued holding while the pulse rate was calculated for 15 s, ranging from 5th–20th second of the retrieval phase. Retrieval heart rate was transformed to be conveyed as beats per minute (15 s heart rate x 4), and the after equation was expressed [15]:Men: VO2 max (mL/kg/min) = 111.33 − (0.42 × heart rate (bpm))
Women: VO2 max (mL/kg/min) = 65.81 − (0.1847 × heart rate (bpm))

Reliability and validity were reported to be *r* = 0.92 and *r* = −0.75, respectively [16].

The 6MWT was accomplished alongside a flat, straight, 40 m track with a hard surface. Each participant was permitted to break and stand, as essential, within the allocated 6 min [17]. At the close of the assignment, the walking distance was determined. RPE is a quantifiable amount of perceived exertion applied to evaluate physical activity intensity; participants stated their perceived level of exertion on a mathematical scale, from 6 (no exertion) to 20 (maximal exertion), after the 6MWT. RPE is an incidental procedure for the estimation of cardiopulmonary condition stamina [18,19] (Figure 1).

BMI is the percentage of a participant’s weight to height squared (kg/m^2^), and it is applied to approximate an individual’s risk of weight-associated physical condition challenges. BMI calculates excess weight for a specific height. It is not a directly measured quantity of body fat. BMI is the highly commonly utilized amount of weight-associated health threat because explicit methods of body fat (e.g., skinfold sizes or underwater weighing) are more aggressive and expensive [20]. A BMI capacity is comparatively simple, cheap, noninvasive as well as fast [21]. Waist circumference is commonly used, but kg/m is used because there is no difference between the same race, gender, and age.

The post-questionnaire included questions on motivation, fun, and the perceived exercise effectiveness section had questions. The scale ranged from 0 (“none”) to 10 (“max”). All participants participated in this survey.

### 2.3. Super Kids Adventure

The SUKIA game provides upper extremity stretching, lunging, boxing, side-bending, squatting, and arm and jumping exercises. Furthermore, the game also increases back muscle strength. On SUKIA, players can selectively play games by selecting various characters. Various characters recognize the player’s movements and move according to the player. Its simple user interface, which includes real-time visual and auditory feedback, was designed to give participants the motivation to exercise. The application has an alarm function that reminds participants to exercise regularly. In SUKIA, gesture recognition is presented employing a convolutional neural network (CNN) in a deep-learning algorithm [22]. Utilizing CNN-based image identification, the characters are moved by applying upper extremity stretching, lunge, boxing, side-bending, squatting, open arm movements, and jumping. The SUKIA CNN was constructed using the first popular algorithm used for exercise and learning multi-layer system network. CNNs are frequently utilized in deep-learning algorithms to distinguish gesture movements (Figure 2).

### 2.4. Intervention

All participants were randomly designated for either the SUIKA or NINS interventions. Each session lasted 30 min and was conducted five times per week for three weeks. However, resting time was provided whenever necessary, though the intervention time consistently lasted for 30 min.

In the SUKIA intervention, participants were instructed to select the SUIKA game after the game was displayed on a smart device and erected using a smartphone holder. Different games were selected every day, and audio feedback was also provided. The number of games played was counted, and video feedback on posture was provided. When participants performed the correct posture for any of the exercises facilitated by the SUIKA games, the character moved and scored (Figure 3).

With respect to the NINS intervention, “Ring Fit Adventure” was employed. NINS is a health-oriented role-playing game that utilizes a ring-shaped controller as a method for endurance training. The participant advances through the narrative even as physical exercise, as the organization of the player is associated with the major fictional character on the computer screen (Figure 4). The switch is provided with a high-accuracy strength sensor that identifies in addition to digitizes the participant’s motion, for instance, squeezing and stretching the switch. Based on this functional feedback, the video game itself approximates the ideal workout strength for each participant and executes excellent tune-up and down-regulation. Thus, it has developed the potential to deliver an applicable measure of training for all age groups. In adventure mode, the participant switches the fictional character by squatting or jogging. In a battle event, including an aerobics mode menu, rigorous endurance exercises and yoga workouts that utilize stress on the whole-body muscles are performed while the player aims to defeat the opponent and clear the game stage. Participants are compensated along with the quantity of training they operate and maintain to progress, constantly enhancing their talents. All participants were asked to participate in the game with an intensity of about 4–6 MET. Participants were asked to participate in the game so hard when they walked at a fast pace. Otherwise, the participant can alter the training set menu for a particular objective, for instance, increasing shoulder stiffness or reducing low back pain, for a little more rigorous exercise training program [23]. In both groups, calorie consumption and RPE were checked every session to calculate the average. After all the sessions, VO2 max, 6-MWT, and BMI were measured.

### 2.5. Statistical Analysis

The table results were confirmed as means and standard deviations (SDs). All variables were analyzed applying the Kolmogorov–Smirnov test, assuming a normal distribution. A power analysis, applying G-Power software, was completed to determine the sample size necessity (N = 20) based on our pilot research paper, which showed effect size (eta squared, η2 = 0.5) and power (1-β = 0.8). Considering the dropout rate of 20%, 24 participants were recruited in this study. Chi-squared tests were used to determine variability across genders. Independent *t*-tests were employed to define demographic data and clinical outcome measurements. Independent *t*-tests were used to calculate calorie consumption, RPE, and post-questionnaire for the motivation, fun, and perceived exercise effectiveness. A two-group, two-intervention, repeated-measures analysis of variance (ANOVA) was managed to discover BMI, VO2 max, and 6MWT. Tukey’s post hoc analysis test was used to assess whether the interaction and main effects were examined. The time effect refers to the difference between before and after intervention. Additionally, SPSS software (version 26.0, SPSS, Chicago, IL, USA) was used to achieve statistical analyses. A *p*-value was set to 0.05.

## 3. Results

### 3.1. Demographic Characteristics of Participants

The demographic attributes of the participants are shown in Table 2. All participants successfully completed the experimental tests and interventions. There were no significant variations in sex, age, height, weight, and BMI between the SUKIA and NINS groups (*p* > 0.05).

### 3.2. Calorie consumption and RPE

The independent *t*-tests showed a significant difference in calorie consumption and RPE between the SUKIA and NINS groups, respectively (*p* = 0.03; 0.04), supporting better calorie consumption and cardiopulmonary endurance following SUKIA for adolescents with obesity (Table 3).

### 3.3. Body Mass Index

The repeated-measures ANOVA showed a significantly time (*p* = 0.001) difference in the BMI data analysis. The paired *t*-tests showed a significantly difference in the BMI between the pre- and post-tests of the SUKIA (*p* = 0.02) and NINS (*p* = 0.03) groups. The independent *t*-tests and Tuckey’s post hoc tests did not reveal significant differences in BMI between both groups, suggesting both interventions were more effective in reducing fat compared to pre-intervention (*p* = 0.471) (Table 4).

### 3.4. VO2 Max and 6-min Walking Test

The repeated-measures ANOVA significantly indicated time (*p* = 0.001) and time x group interaction effects (*p* = 0.005) for VO2max. The paired *t*-test confirmed a significantly difference in VO2 max between the pre- and post-tests of both groups (*p* = 0.001; 0.001). The post hoc test analysis confirmed a greater raise in VO2 max in the SUKIA compared to NINS in adolescents with obesity (*p* = 0.001), suggesting both interventions were more improved in terms of cardiopulmonary function compared to pre intervention. (Table 5).

The repeated-measures ANOVA significantly showed time (*p* = 0.001) and group main effects (*p* = 0.039) for the 6MWT. The post hoc analysis confirmed a greater increase in the 6MWT in the SUKIA than NINS in adolescents with obesity (*p* = 0.01). A paired *t*-test revealed a significantly difference in the 6MWT between the pre- and post-tests of both groups (*p* = 0.02; 0.03). An independent *t*-test revealed a significantly difference in the 6MWT between both groups after intervention (*p =* 0.03; 0.01). Moreover, post hoc test analysis using Tukey’s test indicated a greater raise in 6MWT in SUKIA than NINS in adolescents with obesity (*p* = 0.03) (Table 5).

### 3.5. Post-Questionnaire for the Motivation, Fun, and Perceived Exercise Effectiveness

The independent *t*-test did not show significant difference in motivation, fun, and perceived exercise effectiveness between SUKIA and NINS (*p* = 0.57; 0.34 and 0.28, respectively), suggesting that participants were satisfied with SUKA and NINS (Table 6).

## 4. Discussion

This current randomized clinical trial investigated the therapeutic effects of SUKIA on cardiopulmonary function and endurance, calorie consumption, and BMI in adolescents with obesity. As anticipated, SUKIA reduced weight while improving physical activity in adolescents with obesity. Weight and physical activity changes were more pronounced and thus more favorable in adolescents with obesity. To the greatest of our understanding, no previously published data are available for comparison with our clinical data.

Energy expenditure analysis demonstrated a greater increase (12.4%) in calorie consumption during SUKIA than during NINS. This finding was in parallel with the energy expenditure results. Unnithan and colleagues (2006) observed an increase in calorie expenditure (58.6%) after 12 min of Dance Revolution in 10 overweight children compared to non-overweight children [24]. This finding suggests that home exercise games may have facilitated anxiety lessening in participants with feelings of anxiety, and encouraged stress decline and relaxation, thus helping a subsection of participants with obesity in improving their body enjoyment and self-efficacy. Home exercise games have excellent ability to augment treatment reliability [4]. Even when applying a manualized medication procedure supervised for conformity, individuals demonstrate substantial inconsistency in their supply of behavior mediation. In conflict, receiving teaching provided by a trained virtual character rather than a live character guarantees dependable intervention provision. Chueca and colleagues (2020) reported increased kcal/min (4.28%) when using the Nintendo Wii compared to home exercises in 40 overweight children [23]. This finding suggests that calorie consumption by SUKIA seem to provoke a superior intensity (kcal/min) compared to the NINS, possibly because it involves motion movement of the whole body. Prior research consents with the findings of in this study [24,25]. Most experiments that assessed calorie spending using the Xbox 360 Kinect stated significant differences among girls and boys in kcal paid out, as did this experiment [26,27]. Smallwood and colleagues (2012) observed the energy spend of two games for the Xbox Kinect (Dance Central and Kinect Sports boxing) in adolescents between 11 and 15 years old [28]. The energy spend was 172 kcal/h. In particular, for Kinect Sport boxing (an active video game used in this experiment), the energy spend was 4.4 METs. Canabrava and colleagues (2018) showed similar results regarding the parallel in energy spend between ambulatory capacity at 5 km/h and play-acting using the Xbox Kinect (Kinect Adventures, Kinect Sports boxing, and a dance videogame), showing an energy spend of 185 kcal/h and presentation that active video games can be an exciting alternative to high physical movement and a substitute for conventional games [29]. Once more, there were no considerable changes in energy costs between adolescents with a normal weight and overweight or obese adolescents.

Cardiopulmonary function analysis demonstrated that SUKIA more effectively increased RPE (12.6%) and VO2 max (5.1%) in the SUKIA group than in the NINS group, while there was a greater increase in 6MWT (1.66%) in the NINS group than the SUKIA group. This conclusion mirrors earlier cardiopulmonary results. Juliana and colleagues (2019) observed increased VO2 peak (52.9%) and Borg (26%) after Kinect video games compared to cardiopulmonary exercise tests in 30 asthmatic children [28]. Dickinson and Place reported increased VO2 max (230%) in 100 children with autism after 1 year of playing Nintendo Wii (Mario and Sonics at the Olympics) [30]. These results may support the growing number of application-established programs that approach managed corrected fitness events; however, programs that are active and game based have been shown as being of modest importance to wellbeing and health. This finding suggests that movement recognition applications may be a useful motivational setting for promoting exercise that increases the movement velocity of the whole body, as well as the bilateral use of the extremities and trunk. It could be argued that SUKIA is like interval training; in fact, gaming exercises use a number of different games, each lasting for up to 2–3 min. The SUKIA group improved more in 6MWT compared to the NINS group and contributed to promoting motivation and improving cardiorespiratory function based on the post questionnaire assessing motivation, fun, and perceived exercise effectiveness.

BMI analysis demonstrated that the SUKIA group had a higher decrease in BMI (1.28%) than the NINS group. Adamo and colleagues (2010) observed a reduced BMI (0.3%) in 18 obese adolescents after playing video games, compared to listening to music [31]. Maddison and colleagues (2011) observed a decreased BMI (3.12%) in 231 participants following dynamic video game play, compared to the control group, after 24 weeks [32]. BMI is an empirical measurement of body mass based on the mass and height of an individual. In adolescence, BMI rises as height and body mass increase. In the current research, adolescents in groups together were often a similar height with minimum difference over the research phase; nevertheless, the distinction in weight encouraged the intervention group with 1 kg less mass benefit contrasted with that in the control group. Both dynamic video games should be intended to augment entire body movements, which have a physiological wellbeing benefit. This is associated with the possibility of focusing on recipients of dynamic video game interventions; in adolescents [4,24], the comparatively greater workout strength may give better wellbeing advantages. Technology acknowledgment was calculated by organizing an enjoyable and helpful participant satisfaction questionnaire, for SUKIA as much as NINS, and most participants who successfully completed intervention reported it to be fun, safe, as well as helpful for home exercise teaching. A major limitation is that despite what the current study is suggesting, the results should be understood precisely when struggling to deduce the present decisions for clinical exercise and for the supervision of children using adolescent training, because of the small sample size and limited experiment time. Another limitation is that the effects of the intervention methods alone could not be compared due to the inability to limit diet/work and rest rules. Future research should concentrate on greater, more systematically signaled mediation experiments to give authoritative responses regarding whether this expertise is useful in supporting lasting physical movement in adolescents. Home exercise video gaming knowledge is persistently growing and being refined to adjust to customer needs for innovative gaming capabilities. Mediations that apply this expertise should benefit from this to offer a pleasing intervention to the people in focus.

## 5. Conclusions

Both SUKIA and NINS had improved more in terms of BMI, VO2 max, and 6MWT compared to pre-intervention. SUKIA involving body movements to control characters in the game, as played by obese children, was found to be associated with improved BMI, cardiopulmonary function, calorie consumption, and clinical status. Our results highlight that the incorporation of SUKIA allows AI to freely offer accurate visual feedback as well as effective therapeutic home exercise training, which could serve as a basis for advanced AI-based home exercise science research.

## Figures and Tables

**Figure 1 sensors-22-07352-f001:**
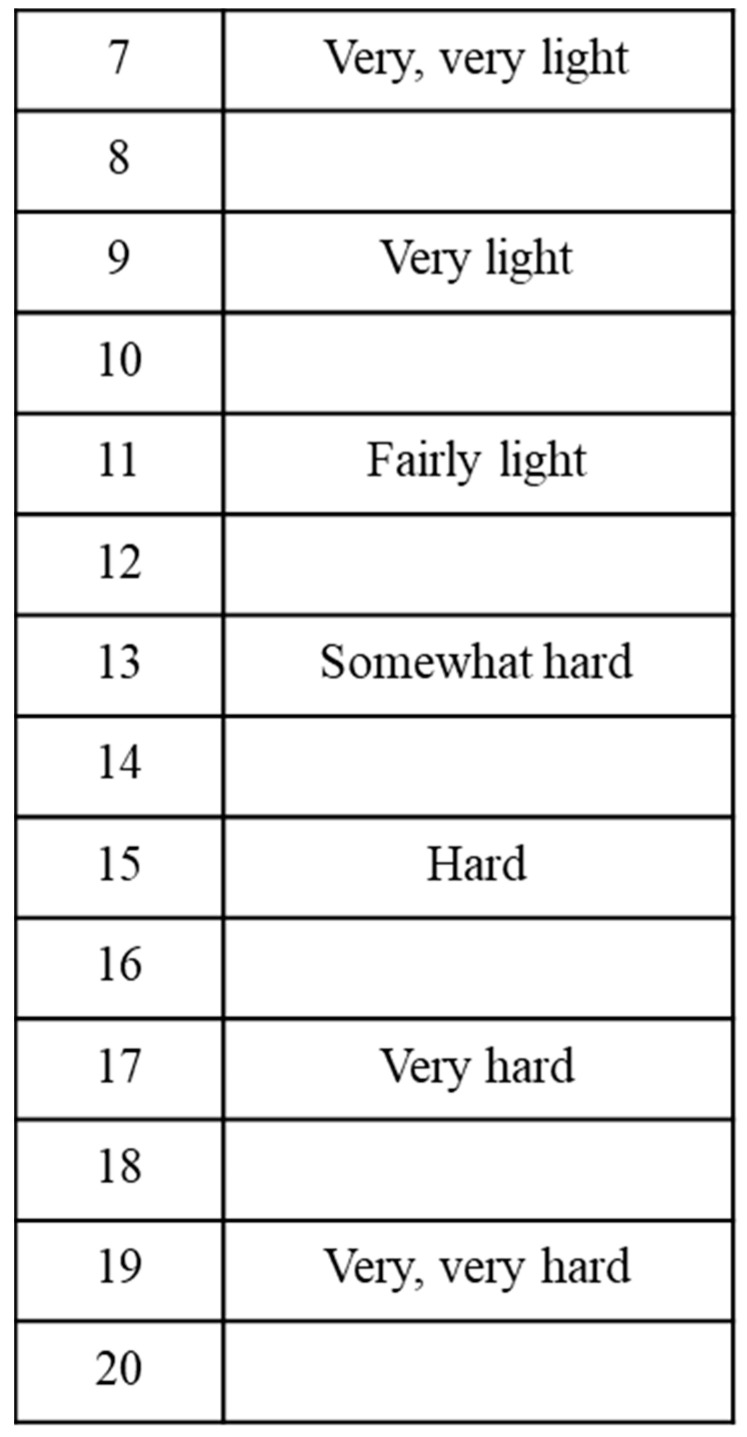
Borg rating of perceived exertion scale.

**Figure 2 sensors-22-07352-f002:**
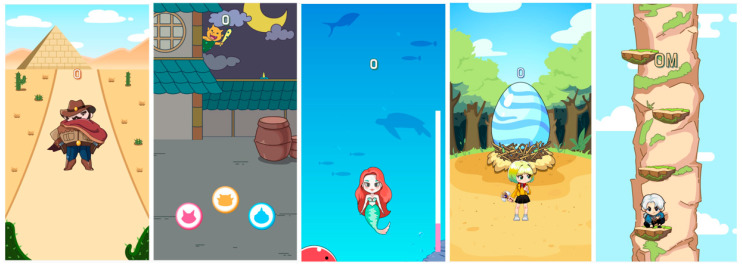
Super Kids Adventure home exercise game.

**Figure 3 sensors-22-07352-f003:**
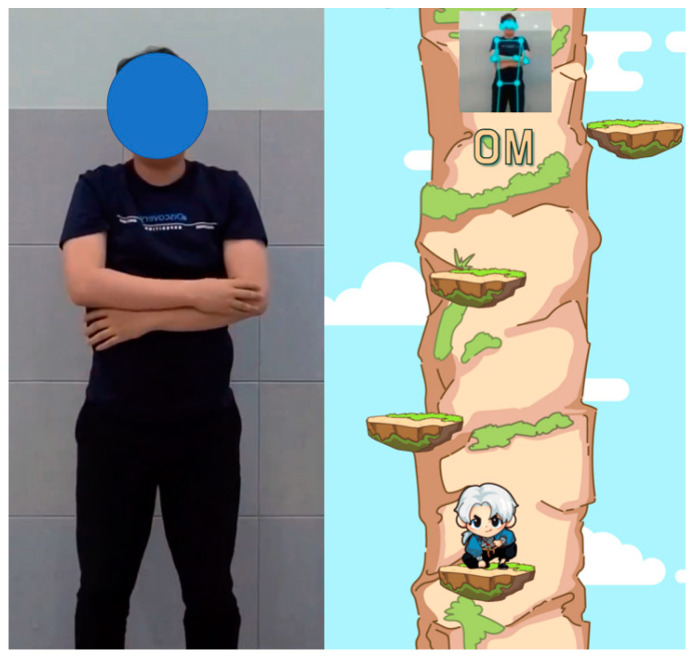
SUKIA intervention.

**Figure 4 sensors-22-07352-f004:**
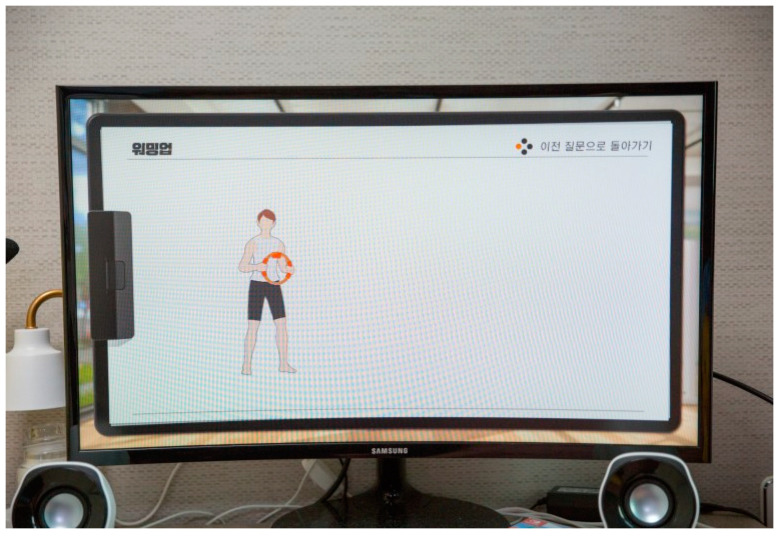
Nintendo switch game screen.

**Table 1 sensors-22-07352-t001:** Metabolic equivalent.

Activity	Speed (λ, km/h)	METs
Stay	-	1.3
Walking	λ ≤ 1.0	2.0
	1.0 < λ ≤ 1.5	2.8
	1.5 < λ ≤ 4.0	3.5
	4.0 < λ ≤ 5.0	5.0
	5.0 < λ ≤ 6.4	6.0
Jogging	6.4 < λ ≤ 7.3	8.0
	7.3 < λ ≤ 8.0	10.0
	8.0 < λ ≤ 9.6	13.5
	9.6 < λ ≤ 11.2	16.0
	11.2 < λ	18.0

**Table 2 sensors-22-07352-t002:** Demographic characteristics of participants.

Characteristics	SUKIA ^a^ (*n* = 12)	NINS ^b^ (*n* = 12)	*p*-Value
Age (years)	13.58 ± 1.44	13.08 ± 2.35	0.08
Gender (%)			
Female	2 (16.67%)	2 (16.67%)	1.000
Male	10 (83.33%)	10 (83.33%)	
Height (cm)	135.75 ± 8.81	136.83 ± 7.55	0.88
Weight (kg)	46.42 ± 7.63	47.17 ± 7.61	0.65
BMI (kg/m^2^)	24.99 ± 1.35	25.03 ± 1.92	0.06

^a^ Super Kids Adventure; ^b^ Nintendo Switch.

**Table 3 sensors-22-07352-t003:** Calorie consumption and RPE between SUKIA and NINS.

	SUKIA ^a^	NINS ^b^	*p*-Value
Calorie consumption	455.75 ± 50.42	405.42 ± 54.29	0.03 *
RPE ^c^	12.58 ± 2.50	11.17 ± 1.75	0.04 *

^a^ Super Kids Adventure; ^b^ Nintendo Switch; ^c^ Borg rating of perceived exertion scale; * *p* < 0.05.

**Table 4 sensors-22-07352-t004:** Body mass index between SUKIA and NINS.

Body Mass Index (kg/m^2^)	SUKIA ^a^	NINS ^b^	*Time Effect*	*Group* *Effect*	*Time x Group Interaction*
Pre	24.99 ± 1.35	25.03 ± 1.92	0.001 *	0.768	0.496
Post	23.93 ± 1.83	24.29 ± 1.83			

^a^ Super Kids Adventure; ^b^ Nintendo switch; * *p* < 0.05.

**Table 5 sensors-22-07352-t005:** VO2 max and 6 min walking test between SUKIA and NINS.

		Pre-Test	Post-Test	*Time Effect*	*Group* *Effect*	*Time x Group Interaction*
VO2 max (mL/kg/min)	SUKIA ^a^	28.85 ± 2.92	30.62 ± 2.98	0.001 *	0.150	0.005 *
	NINS ^b^	28.98 ± 2.35	29.28 ± 2.43			
6MWT (meter)	SUKIA ^a^	521.83 ± 22.72	545.50 ± 22.84	0.001 *	0.039 *	0.361
	NINS ^b^	495.75 ± 31.15	526.42 ± 29.59			

^a^ Super Kids Adventure; ^b^ Nintendo Switch; * *p* < 0.05.

**Table 6 sensors-22-07352-t006:** Post-questionnaire for the motivation, fun, and perceived exercise effectiveness.

	SUKIA ^a^	NINS ^b^	*p*-Value
Motivation	7.80 ± 1.20	7 ± 1.40	0.57
Fun	8.78 ± 0.67	7.93 ± 1.30	0.34
Perceived exercise effectiveness	8.50 ± 0.57	8.00 ± 1.00	0.28

^a^ Super Kids Adventure; ^b^ Nintendo Switch.

## Data Availability

Not applicable.

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
