# Peer review of "Comparative Effectiveness of Artificial Intelligence-Based Interactive Home Exercise Applications in Adolescents with Obesity"

_sensors, 2022, doi:10.3390/s22197352_

Round 1

Reviewer 1 Report

This manuscript describes an artificial intelligence 15 (AI)-based gesture recognition super kids adventure game application (SUKIA, Funrehab, Daejeon, 16 Republic of Korea) for assessing the effects of SUKIA and Nintendo Switch (NINS) on calorie 18 consumption, VO2 max, 6-minute walking test (6MWT) as well as body mass index (BMI), and the 19 Borg rating of perceived exertion scale (RPE) in adolescents with obesity. The results provide evidence that 24 SUKIA can more effectively improve cardiopulmonary function and calorie consumption than 25 NINS in adolescents with obesity during COVID-19. However, more explains are needed to strengthen authors' statements.

1. There may be other life factors that can significantly interfere with the results (diet/work and rest rules), how to control them in the experiment? Did authors consider the individual differences of volunteers? How do avoid sample bias when the number of people is limited and the experiment time is short?

2. What interference did the experimenters do to the volunteers? How did these interferences affect the experimental results?

3. The work involves the comparison between the experimental group and the control group, and many experimental factors cannot be matched one-to-one (for example, the specific form of the game, and how to measure the activities of different games?), in this case, how to conclude that the experimental group is better than the control group?

Author Response

09.19.22

Ms. Aurora Tang

Section Managing Editor

Sensors

Dear Editor:

Please find enclosed our manuscript entitled “Comparative effectiveness of artificial intelligence interactive home exercise applications in adolescents with obesity”, which we request you to consider for publication as an Article in Sensors.

In this study, we focused on aiming to assess the effects of SUKIA and Nintendo Switch (NINS) on calorie consumption, VO2 max, 6-minute walking test (6MWT) as well as body mass index (BMI), and the Borg rating of perceived exertion scale (RPE) in adolescents with obesity. The participants in this study were 24 adolescents with obesity. The present study showed that Analysis of variance (ANOVA) and independent t-tests were presented with significant level at P < 0.05, and the analysis indicated that SUKIA showed superior effects on calorie consumption, VO2 max, and RPE compared to NINS.. This manuscript has not been published elsewhere and is not under consideration by another journal. We have approved the manuscript and agree with submission to Neurorehabilitation. There are no conflicts of interest to declare.

We believe that the findings of this study are relevant to the scope of your journal and will be of interest to its readership. The manuscript has been carefully reviewed by an experienced editor whose first language is English and who specializes in editing papers written by scientists whose native language is not English.

We look forward to hearing from you at your earliest convenience.

Sincerely,

Chanhee Park, PT, PhD

Department of Physical Therapy, 1 Yonseidae-gil, Wonju, Gangwon-do 26493,

Republic of Korea

Email Address: [email protected]

Reviewer 2 Report

The paper is interesting, and it presents some research to measure the differences in energy expenditure using 2 gaming systems. I have some comments to make to the authors regarding this paper that I would like them to address.

The definition of METS is difficult to interpret from what the authors description of a MET is. A MET is recognised as the objective measure of the ratio of the rate at which a person expends energy, relative to the mass of that person, while performing some specific physical activity compared to a reference, set by convention at 3.5 mL of oxygen per kilogram per minute, which is roughly equivalent to the energy expended when sitting quietly. The paper [12] referenced by the authors does not describe the data shown in Table 1. And other relevant previous studies suggest that jogging on level ground at 9 km/h has a METS value of 8.8, increasing to 11.2 when jogging at 11 km/h. See Jetté, M., Sidney, K. and Blümchen, G. (1990), Metabolic equivalents (METS) in exercise testing, exercise prescription, and evaluation of functional capacity. Clin Cardiol, 13: 555-565. https://doi.org/10.1002/clc.4960130809 for further information.

Therefore, I recommend that the authors provide more justification as to the data presented in Table 1. And I suggest that the authors need to examine METS data from several previous studies to arrive at a consensus with their MET to activity data, and then update table 1 accordingly.

The authors used BMI as a metric to calculate the amount of weight-associated health threat that each participant had. Although this is fine, it is a reasonably outdated method of body fat content measurement, and it has been well documented that it does not take into account muscle mass, bone density, overall body composition, and racial and sex differences, and waist circumference is a better method of body fat composition and is commonly used in research studies. Therefore I would like to see a short discussion around the discussion in lies 162 – 171 to describe why waist circumference was not used as a measurement of body fat composition.

The authors state in line 175 that the SUKIA game increases upper extremity and back muscle strength. A reference hasn’t been provided to verify this statement, so I recommend that the authors add a reference to the original source of this information.

Furthermore, the authors claim that “The suggested SUKIA algorithm can decrease pre-processing time and instantly acquire the highly expressive features since the initial data input with not recognizing the physical characteristics”. This discussion seems irrelevant to the main focus of the paper and should be removed.

Section 2.4 is very difficult to read and needs major English editing to increase ease of reading.

The paper needs a section that describes in detail how both measurement systems are comparable. For example, the SUKIA intervention measures activity through postures, and the NINS intervention uses a ring-shaped controller to measure stretching and squeezing, and motion. How was this data extracted out of each game for comparison? Was it simply that both games were set to encourage movement within 4-6 METS? Or did the authors have access to the raw sensor data from both systems? Please explain in detail how data was extracted from both systems, and any settings that were configured on each system to make them both comparable.

Section 2.5 suggests that data from both systems were firstly analysed for normality before independent t-tests were applied to calculate calorie consumption etc. And a two-group, two-intervention, repeated measures analysis of variance (ANOVA) was used on BMI, VO2 max, and 6MWT data. Where did this data come from? See earlier comment. Please explain further.

Tables 4 and 5 show a “Time effect” column. What does time effect refer to? Please explain further how it is calculated and why it is important in this study. Perhaps this will be explained in my previous comments about the need to explain bow both systems were assessed for comparisons.

Table 4 suggests that the BMI of both groups using both systems dropped by approximately 1. Please provide a comment on what this indicates.

Table 5 suggests that participants using SUKIA improved by approx. 24 metres whereas the NINS participants improved by 29 metres. Is this a significant improvement, or could it be caused by other unrelated effects such as lack of motivation or concentration by the participants? Please explain.

Author Response

(The authors gave the same response as above.)

Round 2

Reviewer 1 Report

Corrections to minor methodological errors and text editing.

Author Response

We made corrections for minor methodological errors and text editing.

Reviewer 2 Report

The paper is now better since my suggested improvements have been applied. Therefore I have no further comments to add.

Author Response

Thank you for reviewing my manuscript.

Best Regards

Chanhee